# Performance Evaluation of a Low-Cost Non-Invasive Ventilator during the COVID-19 Pandemic: A Bench Study

**DOI:** 10.3390/healthcare10112229

**Published:** 2022-11-07

**Authors:** Nils Correvon, Jean-Bernard Michotte, Olivier Contal

**Affiliations:** School of Health Sciences (HESAV), HES-SO University of Applied Sciences and Arts of Western Switzerland, 1005 Lausanne, Switzerland

**Keywords:** non-invasive ventilation, bench test, COVID-19, mechanical ventilation

## Abstract

Background: During the COVID-19 breakout, a global call for low-cost portable ventilators was made following the strong demand for ventilatory support techniques. Among a few development projects, COVIDair non-invasive ventilator was developed and produced in a record time during the critical period of spring 2020. Objectives: To evaluate COVIDair performance (i.e., inspiratory trigger delay time, TDT, pressurization time and inspiratory to expiratory time ratio, I:E) on a test bench simulating physiological characteristics of breathing. Method: Performance tests were conducted on a breathing simulator (ASL 5000, IngMar Medical™) in two different lung mechanics (i.e., normal and severe restrictive). Results: Under normal pulmonary mechanics, the inspiratory TDT is on average between 89.0 (±2.1) and 135.0 (±9.7) ms. In a situation of severe restrictive pulmonary mechanics, the inspiratory TDT is on average between 80 (±3.1) and 99.2 (±5.5) ms. Pressurization time to pre-set inspiratory pressure was on average from 234.6 (±5.5) to 318.6 (±1.9) ms. The absolute difference between the actual I:E cycling measure and the pre-set I:E cycling value ranged from 0.1 to 10.7% on average. Conclusion: In normal and severe restrictive pulmonary mechanics scenarios, the performance of COVIDair meets the expected standards for non-invasive ventilators.

## 1. Introduction

During the coronavirus disease of 2019 (COVID-19) pandemic breakout, a global call for low-cost portable ventilators was made following the strong demand for ventilatory support techniques, especially in middle to low-outcome countries [1,2]. Few low-cost non-invasive ventilation (NIV) device development projects have emerged [3], yet these devices must meet a certain number of criteria and technical performance in order to allow optimal ventilation of the patient.

Among these projects, the COVIDair device developed by BCD microtechnique SA (Préverenges, Switzerland) during the critical period of spring 2020, aims to ventilate, in a non-invasive manner, patients with severe respiratory pathologies. With the help of a local non-governmental organization, the device has been used in different hospitals in Nepal, Macedonia, and India. BCD microtechnique SA was granted an Innosuisse cheque, a funding instrument of the Swiss Confederation that helps start-ups and small companies to test innovative ideas in collaboration with a research partner.

In partnership with BCD microtechnique SA, we (Haute École de Santé Vaud, HESAV) evaluated the triggering delay time (TDT) of the inspiratory trigger, the pressurization time, and the performance of the inspiratory to expiratory ratio (I:E) cycling of the COVIDair non-invasive ventilator on a bench test simulating physiological characteristics of breathing.

## 2. Materials and Methods

This bench study was performed at the cardio-respiratory laboratory of HESAV in Lausanne.

The tests were performed using a spontaneous breathing simulator (ASL 5000™, IngMar Medical™, Pittsburgh, PA, USA) which was calibrated before the study in accordance with the manufacturer’s recommendations and requirements. The COVIDair was directly connected to the simulator via a standard 2-m single-limb circuit and an exhalation leak valve (Whisper Swivel™ II, Respironics Inc.™, Murrysville, PA, USA). In order to assess the performance of the device, different scenarios were set using different lung dynamics and ventilation settings.

Two types of respiratory dynamics were simulated, the first as normal and the second one as a severe restrictive type, mimicking an acute respiratory distress syndrome encountered in severe COVID-19 cases (Table 1). In the normal type simulation, the following lung characteristics were set on the simulator [4]: a tracheal resistance of 5 cmH_2_O/L/s, a compliance of 80 mL/cmH_2_O, a respiratory rate of 15 cycles/min and an inspiratory muscle effort of −5 cmH_2_O with a rise time of 30% and a release time of 10%, where the percentage represents duration as a fraction of the respiratory cycle (determined by dividing 60 s by the respiratory rate). The pulmonary characteristics of the severe restrictive type simulation were as follows: a tracheal resistance of 5 cmH_2_O/L/s, a compliance of 20 mL/cmH_2_O, a respiratory rate of 30 and 40 cycles/min, and an inspiratory muscular effort of −15 cmH_2_O with a rise time of 25% and a release time of 25%.

Performance evaluation was carried out in spontaneous timed mode (S/T) with various device settings. In the different scenarios, the inspiratory and expiratory pressurization slopes as well as the respiratory rate were set to a minimum (i.e., 100 ms and 3 cycles/min in order to avoid any interference with spontaneous breathing). Under normal conditions, the expiratory positive airway pressure setting (EPAP) was 5 cmH_2_O and the inspiratory positive airway pressure (IPAP) was 15 cmH_2_O. Two levels of inspiratory triggers and two levels of I:E cycling were tested at 2 and 5 L/min, and 10 and 25%, respectively. In the severely restrictive situation, the ventilator was adjusted in two ways. First, with an EPAP of 5 cmH_2_O, an IPAP of 15 cmH_2_O, an inspiratory trigger at 5 L/min, which do not generate any self-triggers, and I:E cycling of 10 and 25%. Then, in a second step, with an EPAP set at 10 cmH_2_O, an IPAP at 20 cmH_2_O, an inspiratory trigger at 5 L/min (absence of self-triggering) and an I:E cycling at 10%. In the case of self-triggering episodes, the inspiratory trigger was adjusted accordingly to optimize synchronization as it would be in clinical reality.

During the different scenarios, the flow as well as the muscle pressure and airway pressure were computed and recorded continuously for 60 s by the ASL 5000 software. The performance evaluation of COVIDair was performed by measuring the inspiratory TDT (determined by the difference between the start of inspiratory effort, measured on the muscle pressure curve, and the start of pressurization corresponding to the minimum airway pressure, measured on the mouth pressure curve), time of pressurization (determined by the difference between the onset of pressurization and when the measured airway pressure reaches the pre-set pressure (i.e., IPAP setting on the NIV device), as well as the actual measure of the I:E cycling corresponding to the effective ratio between the measurement of the peak expiratory flow (PEF) and the flow measured at the end of the pressurization of the device, corresponding to the release of the pressurization curve [4]. Using the ASL 5000 Post-Run Analysis software, these measurements were carried out manually over the five respiratory cycles following the 20th second of the various scenarios.

## 3. Results

In total, eight scenarios were tested to assess the performance of COVIDair: four in the context of normal pulmonary mechanics and four in the context of severe restrictive pulmonary mechanics that can be found in severe cases of COVID-19. Firstly, it can be mentioned that for all the conditions evaluated, the device reached the prescribed pressure. Results are expressed as mean (±standard deviation, SD).

Under normal pulmonary mechanics, the inspiratory TDT was on average between 89.0 (±2.1) and 135.0 (±9.7) ms. In a situation of severe restrictive pulmonary mechanics, the inspiratory TDT was on average between 80 (±3.1) and 99.2 (±5.5) ms. The details of the inspiratory TDT are in Table 2.

In severe restrictive type scenarios, self-triggering events were observed when EPAP and IPAP were pre-set at 10 and 20 cmH_2_O respectively and the inspiratory trigger at 5 L/min. When decreasing the sensitivity of the inspiratory trigger down to 7 L/min during these two scenarios, no auto-triggering was observed.

Pressurization time to pre-set IPAP was slightly faster at higher pressure levels with, on average, a pressurization time of 234.6 (±5.5) to 250.6 (±2.5) ms at EPAP/IPAP at 10/20 cmH_2_O versus 298.8 (±6.5) to 318.6 (±1.9) ms at EPAP/IPAP levels at 5/15 cmH_2_O (Table 3).

The absolute difference between the actual I:E cycling measure and the pre-set I:E cycling value ranged from 0.1 to 10.7% on average. In the situations of severe restrictive pulmonary mechanics with EPAP/IPAP setting at 5/15 cmH_2_O, it was not possible to observe a plateau pressure, hence to define the end of pressurization (Table 4).

## 4. Discussion

In normal pulmonary mechanics scenarios, the performance of COVIDair meets the expected standards for non-invasive ventilators. For example, when the inspiratory trigger sensitivity is optimally set, the TDT observed in this study is better than those obtained by Chen et al. (2015), with average values greater than 120 ms. It should be mentioned that no issues with auto-triggering or asynchrony were observed during the different tests. The values measured for the pressurization time and the I:E cycling performance are comparable to the values in the Chen et al. study (2015) [5].

In severe restrictive pulmonary mechanics scenarios (i.e., COVID-19), the performance of COVIDair regarding inspiratory trigger sensitivity remains the same. With optimal setting, the inspiratory TDT was always less than 100 ms. A scoring system has been proposed to evaluate the performance of ICU ventilators in the context of the COVID-19 pandemic [6]. Even though the investigation was carried out slightly differently in the present study, the performance of the COVID air demonstrated values of inspiratory TDT (86.4 (±2.9) ms) close to the targeted best value (i.e., maximal score in the scoring system) of 60.0 ms in low compliance plus increased respiratory effort COVID-19 scenario (i.e., compliance 20 mL/cmH_2_O, respiratory rate 40 cycles/min and inspiratory effort of 10 cmH_2_O). In the same scenario close to the COVIDair pressurization time, 250.6 (±2.5) ms, was also closer to the best-targeted value in a 100.0 to 900.0 ms range.

Except with a simulated respiratory rate of 40 breaths per minute, no auto-triggering was observed with the tested settings which might be an advantage in the management of severely restrictive patients. In the COVID-19 scenario with a respiratory rate of 40 breaths per minute, auto-triggering asynchronies were observed and could be easily resolved by decreasing the inspiratory trigger sensitivity to 7 L/min. We carried out a comparison with the two most common home NIV devices, an A40™ (Philips-Respironics™, Murrysville, PA, USA) and Stellar™ 150 (ResMed™, San Diego, CA, USA). At a preset pressure level of EPAP/IPAP of 5/15 cmH_2_O and an inspiratory trigger of 3 L/min, COVIDair performed similarly to the A40™, while the Stellar 150™ had continuous auto-triggering asynchronies as shown in Figure 1. However, there is a limitation on the pressurization time which can reach 300 ms. This emphasizes the good performance of the COVIDair in severe restrictive pulmonary mechanism breathing type.

It is important to acknowledge that the experiment was performed in a standardized environment and did not take into account real-life factors that can alter ventilation performance. One of the major problems with NIV is the compensation for unintentional leaks and future evaluations should assess this aspect.

## 5. Conclusions

Without prior experience in manufacturing ventilators, BCD microtechnique SA was able to develop and produce an NIV device which fulfills performance standards in a record time. These standards were also met when severe restrictive pulmonary mechanism was simulated.

## Figures and Tables

**Figure 1 healthcare-10-02229-f001:**
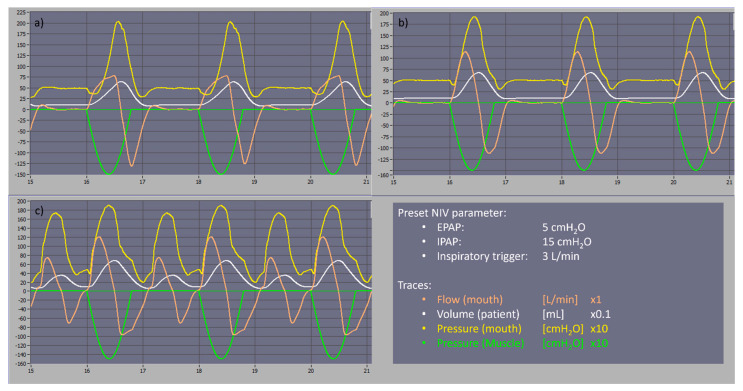
Asynchrony comparison in severe restrictive type scenarios between two the most common NIV devices. Keys: (**a**) COVIDair, (**b**) A40™, (**c**) Stellar™ 150, NIV: non-invasive ventilation, EPAP: expiratory positive airway pressure, IPAP: inspiratory positive airway pressure.

**Table 1 healthcare-10-02229-t001:** Evaluated characteristics.

Scenarios	Pulmonary Mechanic	EPAP/IPAP, cmH_2_O	I:E Cycling, %	Inspiratory Trigger, L/min
1	Normal (RR 15)	5/15	25	2
2	Normal (RR 15)	5/15	25	5
3	Normal (RR 15)	5/15	10	2
4	Normal (RR 15)	5/15	10	5
5	COVID-19 (RR 30)	5/15	10	5
6	COVID-19 (RR 30)	5/15	25	5
7	COVID-19 (RR 30)	10/20	10	7
8	COVID-19 (RR 40)	10/20	10	7

Keys: EPAP: expiratory positive airway pressure, IPAP: inspiratory positive airway pressure, I:E: inspiratory to expiratory ratio, RR: respiratory rate.

**Table 2 healthcare-10-02229-t002:** Inspiratory trigger delay time.

Pulmonary Mechanic	EPAP/IPAP, cmH_2_O	I:E Cycling, %	Inspiratory Trigger, L/min	Mean Inspiratory TDT (SD), ms
Normal	5/15	25	2	90.2 (9.7)
			5	130.8 (15.8)
		10	2	89.0 (2.1)
			5	135.0 (9.7)
COVID-19 (RR 30)	5/15	10	5	99.2 (5.5)
		25	5	91.8 (6.3)
	10/20	10	7	80.0 (3.1)
COVID-19 (RR 40)	10/20	10	7	86.4 (2.9)

Keys: EPAP: expiratory positive airway pressure, IPAP: inspiratory positive airway pressure, I:E: inspiratory to expiratory ratio, TDT: trigger delay time, RR: respiratory rate.

**Table 3 healthcare-10-02229-t003:** Pressurization time.

Pulmonary Mechanic	EPAP/IPAP, cmH_2_O	I:E Cycling, %	Inspiratory Trigger, L/min	Mean Pressurization Time (SD), ms
Normal	5/15	25	5	318.6 (1.9)
COVID-19 (RR 30)	5/15	10	5	298.8 (6.5)
	10/20	10	7	234.6 (5.5)
COVID-19 (RR 40)	10/20	10	7	250.6 (2.5)

Keys: EPAP: expiratory positive airway pressure, IPAP: inspiratory positive airway pressure, I:E: inspiratory to expiratory ratio, RR: respiratory rate.

**Table 4 healthcare-10-02229-t004:** I:E cycling.

Pulmonary Mechanic	EPAP/IPAP, cmH_2_O	I:E Cycling, %	Inspiratory Trigger, L/min	Measured I:E Cycling (SD), % PIF
Normal	5/15	25	2	22.6 (0.2)
			5	24.9 (0.6)
		10	2	5.3 (1.7)
			5	7.3 (0.9)
COVID-19 (RR 30)	5/15	10	5	N.A.
		25	5	N.A.
	10/20	10	7	15.5 (2.2)
COVID-19 (RR 40)	10/20	10	7	−0.7 (4.6)

Keys: EPAP: expiratory positive airway pressure, IPAP: inspiratory positive airway pressure, I:E: inspiratory to expiratory ratio, PIF: Peak Inspiratory Flow, RR: respiratory rate, N.A.: nonapplicable.

## Data Availability

The data presented in this study are available on request from the corresponding author. The data are not publicly available due to privacy restriction.

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
