# Peer review of "Performance Evaluation of a Low-Cost Non-Invasive Ventilator during the COVID-19 Pandemic: A Bench Study"

_healthcare, 2022, doi:10.3390/healthcare10112229_

Round 1

Reviewer 1 Report

The manuscript is clear and concise. The developed device seems promising, however the notion of ventilation performance criteria should be better defined in my opinion. 

All experimental parameters are reported, which ensures the reproducibility of the experiment. However, the methods are unclear at the scenario level and some information is inconsistent: the inspiratory trigger is reported at 5 L/mi in the body of the text while it is at 7 L/min in the table for the last two cases. 

In the results section, I could not find table 2? 

Regarding the I:E ratio, using a threshold (90% of the set high pressure for example) to identify the cycling time would allow to get values for all cases, even if the curve does not show a clear plateau. 

The discussion is interesting and puts the results obtained in perspective with a study of the literature. However, the sentence concerning the pressurization time is not clear. 

In the figures comparing the 3 NIV devices, the flow curve of the COVIDair seems different and the inspiratory peak is lower (80 L/min) compared to the 2 other machines (peak flow closer to 110 L/min). Can you explain this difference?

From the point of view of writing: harmonizing the verbs in the past tense (especially in the results part) would be a plus, and some typos need to be corrected (introduction). 

Reviewer 2 Report

In this study, the authors evaluated on a bench test the performance of a low cost NIV device on some important parameters in ventilation: trigger delay, pressurisation time, and cycling. The authors show modest but clinically acceptable performance in ventilatory or restrictive mechanical situations. The conditions we experienced during the covid pandemic saw the development of such devices in response to a strong clinical need. There is an important need for a third-party evaluation of such devices. In this respect, the rationale for this work is good. Nevertheless, the method of evaluation, and especially the interpretation of the data collected, appears questionable.

Major Comments

-          Although the parameters assessed are important parameters in ventilation, we do not have information on a simple criterion: does the device deliver the prescribed pressure? Specifically, the authors state in the results that: “in the situations of severe restrictive pulmonary mechanics [...] it was not possible to observe a plateau pressure”. In my opinion, this is a major issue, which deserves to be discussed.

-          The situations evaluated did not seem to challenge the device very much:

o   the trigger had to be increased to 7 l/min in some situations to avoid self-triggering. This needs to be discussed and compared to clinical practice. Isn't auto-triggering a common problem in NIV that the authors could have wanted to evaluate?

o   A particularly important issue in NIV is the management of non-intentional leaks. It would have been interesting to evaluate this issue, or at least have it discussed before concluding that the device fully meets the expectations of a NIV device.

-          Pressurization time was set at 100 ms and reached 230 to 320 ms during the experiments. This should be acknowledged as a limitation.

-          All in all, the performance of this device was evaluated under low chalenging conditions and was quite poor. However, the authors conclude that the device meets the expected standards, that it does not cause auto-triggers (in the meantime the trigger had to be adjusted to the lowest setting!) and that its performance is excellent. In my opinion, the results presented are not at all in line with this conclusion.

Minor Comments

-          It would be interesting if the introduction allowed the reader to form a more precise opinion of the actual use made of this device (number of centres involved, number of units manufactured/sold, number of patients treated...)

-          It would have been interesting in the method to give a little more information on how the device works. Is there a calibration process?

-          Methods:

o   Rise time 30%; release time 10%. Please be more specific (30% of?).

o   Where were the pressure and flow recorded? (before/after intentional leak?)

Round 2

Reviewer 2 Report

Most of my comments have not been taken into account.

For example, the authors did not consider discussing the increase required for the inspiratory trigger, or discussing the lack of assessment in the presence of unintentional leaks. One of the major issues in NIV is the compensation of unintentional leaks. This has not been assessed, and it is essential to state this in the discussion before claiming that the device is excellent.

I suggested that to improve the reader's understanding of the method, it should be made clear whether there is a calibration process for the device, what the %ages set for rise time and release time correspond to, and where the recorded flows and pressures were measured. This has not been included in the revised version.

I also suggested that the “excellence” of the device be tempered. As mentioned by the authors, their purpose was not to compare the device to the A40 or the Stellar, however, an important message of their discussion is based on this comparison (“this emphasises the excellent performance of the COVIDair”), even though these additional manipulations are not presented in the methods.

Minor comments:

-          The added Table 2 is called Table 1 and we do not see the column headings

-          The caption of figure 1 seems to be incorrectly written "between most two the most common NIV devices".
